# 1-methylnicotinamide attenuated inflammation and regulated flora in Necrotizing enterocolitis

**Lian Hou, Junbao Du, Jinxin Li, Qing Luo, Quan Kang** [iD] *

Stem Cell Biology and Therapy Laboratory, The Children's Hospital of Chongqing Medical University, Chongqing, China

\* 564799351@qq.com

## Abstract

### Background

Necrotizing enterocolitis(NEC) is a prevalent and destructive illness in neonates. Nicotinamide N-methyltransferase (NNMT) and its derivative, 1-methylnicotinamide (1-MNA), are known to be significant in conditions such as cardiovascular inflammation and renal tubular damage, and 1-MNA has been recognized for its anti-inflammatory effects in various diseases. However, the involvement of NNMT and 1-MNA in the development of NEC remains unclear.

### Methods

We collected intestinal tissues and blood samples from children with NEC and control subjects for biochemical analysis. The NEC rats were induced by hypoxic cold stimulation and lipopolysaccharide, and control, NEC and NEC + 1-MNA groups were established. Neonatal rats were executed on the fourth day and blood, intestinal and fecal specimens were taken for subsequent testing.

### Results

Elevated NNMT and 1-MNA were found in NEC children and NEC rats. Exogenous supplementation of 1-MNA to NEC rats reduced mortality, pathological and inflammatory damage, and inhibited activation of the TLR4-NF-κB pathway in neonatal rats. In addition, 1-MNA improved intestinal barrier function and modulated intestinal flora in NEC rats.

### Conclusion

1-MNA attenuated NEC injury by seemingly inhibiting the TLR4-NF-κB pathway, improving intestinal barrier function and modulating intestinal flora. These findings suggest a potential therapeutic role for 1-MNA in NEC management.

**Data availability statement:** The RNA sequence data used in this study are available from the BioProject database(ID 925809 - BioProject - NCBI). Specifically, the PRJNA925809 transcriptome dataset can be accessed at the following URL: https://www.ncbi.nlm.nih.gov/bioproject/. The dataset includes 4 intestinal tissues from NEC patients, 3 adjacent normal tissues, and 5 normal controls, as well as 3 columns of intestinal tissues from NEC rats and 3 columns of intestinal tissues from normal neonatal rats. The data were preprocessed and normalized using the package Affy in R (version 1. 64. 0) and analyzed using the Limma package in R (version 3. 42. 2). P values were corrected using the Bonferroni-Holm (BH) method, with adjusted P < 0. 05 and log2 fold change > 1 indicating statistically significant genetic differences. If you encounter any issues accessing the dataset, such as the link not working, it may be due to network issues or the link itself. Please ensure the link is correct and try accessing it again. If the problem persists, you may need to contact the NCBI support team for further assistance. contact address: 8600 Rockville Pike, Bethesda, MD 20894. telephone number: (301) 496-2475. e-mail address: bioprojecthelp@ncbi.nlm.nih.gov All other relevant data, including human tissue and blood samples, animal model data, histologic assessments, Western blot analysis, intestinal permeability assay, electron microscopy, ELISA, immunohistochemistry, immunofluorescence, and microbiota analysis, are included within the paper and its Supporting Information files. These data were collected and analyzed as described in the methods section, and the results are presented in the paper. For any additional data or specific inquiries, please contact the corresponding author of the study.

**Funding:** The author(s) received no specific funding for this work.

**Competing interests:** The authors have declared that no competing interests exist.

## Introduction

Necrotizing enterocolitis(NEC) is a widespread and severe disease that impacts newborns, particularly premature babies, making it the leading fatality associated with gastrointestinal complications in this at-risk group [1, 2]. NEC is an important risk factor for colorectal cancer [3]. The global prevalence of NEC is about 7%, with a particular prevalence in very low birth weight neonates weighing less than 1500g [4]. NEC's development stems from multiple critical elements, including immature small intestines, hypoxia, a disrupted intestinal barrier, decreased blood supply to the intestines, and abnormal bacterial colonization in the intestinal tract [5–7]. A key aspect of NEC's development is the triggering of various signaling pathways, with the lipopolysaccharide receptor known as toll-like receptor 4 (TLR4) being particularly influential. Bacterial and inflammatory stimuli activate TLR4, initiating downstream signaling pathways [8, 9].

Emerging studies have underscored the involvement of Nicotinamide Adenine Dinucleotide (NAD) metabolism in the inflammatory reactions, where NAD+ derivatives like nicotinamide (NAM) are crucial in managing the metabolic aspects of inflammation. This process involves the conversion of NAM to nicotinamide mononucleotide (NMN) and back to NAD+, significantly impacting inflammatory conditions like intestinal inflammation and diabetic nephropathy [10–12]. Our analysis of the NAD metabolic pathway revealed elevated levels of Nicotinamide N-methyltransferase (NNMT) in both clinical NEC specimens and NEC rat models. NNMT, a cytoplasmic methyltransferase that catalyzes the N-methylation of NAM to form 1-methylnicotinamide (1-MNA), has been observed in higher concentrations in conditions like cardiovascular and renal tubular injuries, pulmonary arterial hypertension, and has been shown to attenuate inflammation through its metabolite, 1-MNA [13–15]. NNMT overexpression in gastrointestinal cancers enhances cell viability, proliferation, migration, and invasiveness [16]. Moreover, 1-MNA has independently been shown to reduce inflammatory factors and decrease the activation of the NLR family pyrin domain containing 3 (NLRP3) inflammasome and NF-κB [17,18] . Additionally, 1-MNA has shown to enhance physical endurance and alleviate exhaustion in COVID-19-afflicted individuals [19]、 reducing cellular oxidative stress and cell death [20]、 exhibiting anti-thrombotic properties [21]and modulating immunity [22]. However, studies on the role of NNMT and 1-MNA in NEC are lacking. Consequently, we explored the possible roles of NNMT and its metabolite, 1-MNA, in NEC's progression.

In our study, we discovered that 1-MNA declined mortality and histopathological damage in NEC rats, decreased the concentration of inflammatory factors, enhanced intestinal barrier function, suppressed the TLR4-NF-κB pathway's inflammatory response, and adjusted the intestinal microbiota. These results offer fresh perspectives for utilizing 1-MNA in NEC therapy.

## Materials and methods

### Microarray data

We accessed the transcriptome dataset from the BioProject database (ID 925809 - BioProject - NCBI) [23]. The PRJNA925809 transcriptome dataset contained 4 intestinal tissues from NEC patients, 3 adjacent normal tissues and 5 normal controls as

well as 3 columns of intestinal tissues from NEC rats and 3 columns of intestinal tissues from normal neonatal rats. Data were preprocessed to expression values and normalized to verify their quality using the package Affy in R (version 1. 64. 0). Samples from the NEC and normal groups were analyzed utilizing the Limma package in R (version 3. 42. 2) [24]. P values were corrected using the Bonferroni-Holm (BH) method. Adjusted $P < 0.05$ and log2 foldchange $> 1$ indicate statistically significant genetic differences.

## Human tissues and blood samples

Intestinal tissues for the NEC group were collected from babies at Bell stage III, necessitating surgical treatment, while control group tissues were derived from similarly aged babies who suffered from congenital intestinal atresia had undergone intestinal resection. Blood samples for the NEC group were extracted during necessary blood tests for infants diagnosed with NEC, whereas samples for the control group were gathered from infants receiving blood tests for non-inflammatory disorders. The research was sanctioned by the Ethics Committee of Children's Hospital of Chongqing Medical University. Adherence to their guidelines and regulations was maintained throughout the study's procedures. Written informed consent was obtained from parents of infants participating in this study. the day, month and year of the start and end of the recruitment period for this study: 2025/01/13–2025/01/20.

## NEC neonatal rats modeling and pharmacologic intervention

The Animal Ethics Committee of Chongqing Medical University provided clearance for all experiments conducted during our study. Three- to five-day-old SD neonatal rats were categorized into three groups: a Control group, a NEC group, and a NEC + 1-MNA group. NEC modeling references previous studies [25]. SD neonatal rats were taken away from their mothers and fed hypertonic formula via gavage every four hours, along with lipopolysaccharide (Sigma-Aldrich, L2880) at a dosage of 5 mg/kg daily from the commencement of the experiment. Hypoxia and cold stimulation were induced using 100% N2 for 90 seconds, followed by a 10-minute exposure at 4°C twice a day. The NEC + 1-MNA group received 1-MNA (APExBIO, C6682) at a dosage of 100 mg/kg/day, concurrent with NEC induction, following dosages referenced from existing literature [13, 14]. The control group was not given any treatment and was left to be fed by the mothers. The experiment spanned three days, after which the general health and survival rates of the neonatal rats were monitored daily. On the fourth day, after an 8-hour fast, neonatal rats were executed by decapitation, and their intestinal tissues, blood, and feces were harvested for further analysis. The animals were anesthetized with bromoethanol before euthanasia to relieve suffering.

## Histologic assessment of the intestinal tract

The distal ileum 1 cm of intestinal tissue was preserved in a 4% paraformaldehyde solution throughout the night. The samples underwent dehydration, were encased in paraffin, and sliced into 4-micrometer sections. Afterwards, the sections were treated with hematoxylin and eosin stains and permanently sealed for imaging (TEKSQRAY, SQS-40P). Two pathology professionals were invited to perform intestinal pathology assessment without knowledge of sections information, and intestinal histopathology scores determined based on the Nadler criteria [26]. Criteria are as follows: Score 0 (normal): normal intestinal structure, intestinal mucosal villi are intact and undamaged. Score 1: minimal detachment of the submucosa and/or lamina propria. Score 2: moderate detachment of the submucosa and/or lamina propria, along with possible edema in these areas. Score 3: severe detachment of the submucosa and/or lamina propria, accompanied by serious edema, and partial villi loss. Score 4: villi loss and presence of necrosis. Rats that scored 2 or higher on the pathological damage scale were diagnosed with NEC.

## Western blot analysis

Intestinal samples were treated with RIPA buffer (Beyotime, P0013B) while on ice, and the lysates were obtained by centrifugation at 15000 rpm for 15 minutes in a 4°C environment. Protein concentrations were then measured by the

BCA protein assay (Beyotime, P0010), and equal volumes (20 µg) of protein lysates per sample were separated by SDS-PAGE. The separated proteins were then transferred onto a PVDF membrane (Millipore, USA). The membrane was closed with BSA and then incubated with the following primary antibodies:Anti-NNMT (proteintech group, A18033), Anti-TLR4(proteintech group, 19811–1-AP), Anti-Phospho-IκBα(abclonal, AP0999), Anti-IκBα (abclonal, A23223), Anti-β-actin(HUABIO, EM21002), Anti-β-tubulin(Epizyme Biomedical, dLF203). The membranes were subsequently treated with a peroxidase (HRP)-linked secondary antibody (Epizyme Biomedical, LF102) that matched the species. Protein expression levels were observed under a visualizer(iBright CL1500) using chemiluminescent HRP developer (BIO-Rad, USA). Antibody dilution concentrations refer to S1 Table. The original image with molecular weight labeling can be found in S4, S5, S6 and S7.

### Intestinal permeability assay

On day four, after an 8-hour fast, neonatal rats were administered FITC-dextran (Sigma-Aldrich, 46946) via gavage at a dose of 40 mg/100g. Following a 4-hour period post-procedure, peripheral blood was collected, coagulated, and centrifuged at 4000 × rpm for 10 minutes at 4°C. Multiple dilutions of FITC-dextran were used to make labeled curves. 25 ul of serum was mixed with 100 ul PBS and the samples were assayed for fluorescence intensity using an enzyme meter using excitation and emission wavelengths set at 480 nm and 520 nm, respectively. Sample concentrations were calculated from the standardized curves.

### Electron microscopy

Ileal tissue samples were trimmed into 1×1 mm pieces, fixed in 2. 5% glutaraldehyde, and refrigerated at 4°C. The samples were fixed, dehydrated and embedded. Embedded samples were cut into ultrathin sections of 60-70 nm on the ultra microtome(Leica UC7). Following staining with a 2% saturated alcohol solution of uranium acetate and lead citrate solution, the sections were examined with a transmission electron microscope (HITACHI, HT7800) and images were acquired for further analysis.

### Enzyme-linked immunosorbent assay (ELISA)

Blood samples from neonates and neonatal rats were prepared by coagulation followed by centrifugation at 4000 × rpm for 10 minutes in a 4°C environment. Rat ileum tissues were homogenized in PBS (1:9 ratio), centrifuged at 4°C, and supernatants were collected. Levels of 1-MNA, TNFα, and IL1β were quantified using ELISA kits(Meimian, 4504/17106).

### Immunohistochemistry

Tissue sections from all groups underwent dewaxing, hydration, and antigen retrieval in EDTA buffer. Once the blocking step was complete, sections were incubated with Anti-Occludin (abclonal, A24601), Anti-ZO-1 (abclonal, A11417), and Anti-Claudin 1 (ZEN-BIOSCIENCE,
    343203), respectively, at 4°C overnight, followed by species-specific secondary antibodies. Finally, the sections were treated with DAB stain, while the nuclei were treated with hematoxylin for 4 minutes. After dehydration and sealing, images were captured using a slide scanning device(TEKSQRAY, SQS-40P). Antibody dilution concentrations refer to S1 Table.

### Immunofluorescence

Sections from three groups were processed for dewaxing, hydration, and antigen retrieval. They were blocked and incubated with Anti-TLR4 (proteintech group, 19811–1-AP), anti-Phospho-IκBα (abclonal, AP0999), and anti-IκBα (abclonal, A23223), respectively. After incubation with corresponding secondary antibodies, cells were re-stained with DAPI for nuclear visualization. Finally, images were scanned and collected under a fluorescent confocal microscope(KFBIO, KF-FL-020). Antibody dilution concentrations refer to S1 Table.

                                                                                           

## Microbiota analysis

Colon feces were collected from neonatal rats with NEC and immediately placed into 1. 5 ml sterile tubes. The samples were then placed in liquid nitrogen for subsequent amplification and sequencing. Then genomic DNA was extracted. The V3-V4 highly variable region of the bacterial 16S rRNA gene was targeted for amplification using primers 341F (CCTAY-GGGRBGCASCAG) and 806R (GGACTACNNGGGGTATCTAAT) on the extracted DNA. The PCR protocol involved an initial denaturation step at 98°C for one minute, then 30 cycles consisting of denaturation at 98°C for 10 seconds, annealing at 50°C for 30 seconds, and extension at 72°C for 30 seconds, with a final extension phase at 72°C for five minutes. To quantify and characterize the PCR products, equivalent amounts of 1X loading buffer, (containing SYBR Green)were blended with the PCR products, and electrophoresis was conducted on a 2% agarose gel for detection. Specimens exhibiting a prominent band within the range of 400–450 base pairs were selected for subsequent experimentation. The PCR product mixtures were then purified utilizing Qiagen Gel Extraction Kit (Qiagen, 28704). Library preparation and sequencing:DNA fragments' paired-end sequences were consolidated using FLASH, a tool that unites paired-end sequences, particularly when there is an overlap between the reads from complementary DNA strand ends. These paired-end sequences were then allocated to their respective samples based on distinctive barcodes. Finally, raw letter analysis using QIIME2 software yielded ASV cluster, species annotation, Community composition, α-diversity and β-diversity.

## Statistical analysis

Statistical analysis was conducted with GraphPad Prism 8. 0. Data are depicted as mean ± SD for normally distributed data. Significance was determined using t-tests for two-group comparisons and one-way ANOVA for analyses involving three or more groups ($P < 0. 05$). For data that did not follow a normal distribution, are described using median and interquartile range, with differences assessed using the Kruskal-Wallis test ($P < 0. 05$).

## Results

### 1. Elevated NNMT and its metabolite 1-MNA in NEC

Bioinformatic analysis of RNA sequencing results revealed elevated NNMT expression in NEC rats (Fig 1A) and NEC children (Fig 1B). To validate the results, western blot was used to assess NNMT protein expression levels, which were significantly elevated in in NEC rats (Figs 1C and 1E) and NEC children (Figs 1D and 1F). Additionally, ELISA was employed to measure 1-MNA concentrations in blood. The results showed that 1-MNA concentrations were elevated in the blood of both NEC rats (Fig 1G) and NEC children (Fig 1H).

### 2. 1-MNA attenuated NEC Rat mortality, injury and inflammation

Survival analysis (Fig 2A) demonstrated significant differences among the control group, NEC group, and NEC + 1-MNA intervention group ($P < 0. 05$). However, 1-MNA intervention have No significant effect the body weight of neonatal rats (Fig 2B). Visual inspection of the intestines from neonatal rats showed significant pneumatosis and hemorrhage in the NEC group, which was attenuated by 1-MNA intervention and not observed in the control group (Fig 2C). Hematoxylin and eosin (H&E) staining (Fig 2D) showed that the intestinal tissues from the control group maintained structural integrity with no edema or detachment of villi and a thicker, intact muscle layer. In contrast, the NEC group exhibited serious edema and detachment of villi, with thinning and detachment of the muscularis propria. This was significantly attenuated by 1-MNA intervention, which resulted in relatively normal gut morphology. Representative images for each pathology score are referred to S2 File. Histopathologic scoring revealed statistically striking differences among the three groups, with the NEC + 1-MNA group scoring markedly lower than the NEC group (Fig 2E). ELISA detection of TNF-α and IL-1β in intestinal tissues demonstrated significantly elevated levels in the NEC group, which were reduced in the NEC + 1-MNA group (Figs 2F and 2E).

**Fig 1. Elevated NNMT and its metabolite 1-MNA in NEC.** (A) Heatmap showing differential expression of NAD metabolic pathways in NEC rats and controls. (B) Heatmap showing differential expression of NAD metabolic pathways in NEC children and controls. (C) western blot to determine NNMT protein expression levels in NEC rat and controls. (D) western blot to determine NNMT protein expression levels in NEC clinical specimens and controls. (E) Relative expression levels of NNMT in NEC rats and controls. (F) Relative expression levels of NNMT in NEC clinical specimens and controls. (G) Elisa assay for 1-MNA concentration in blood of NEC rats and controls. (H) Elisa assay for 1-MNA concentration in blood of children with NEC and controls.

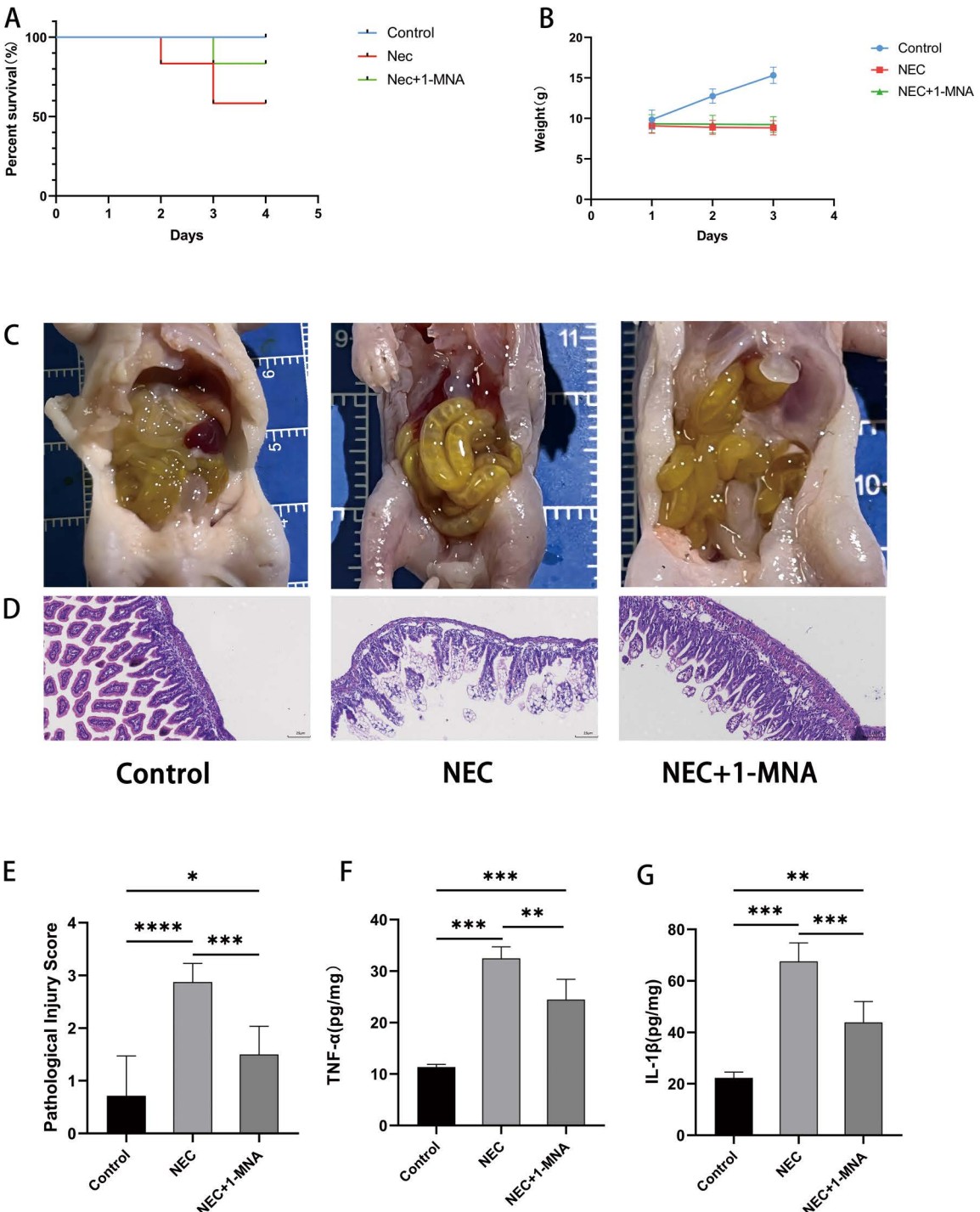

**Fig 2. 1. Effects of MNA on general condition, pathomorphology and inflammatory factors in NEC rats.** (A) Survival curves of neonatal rats. (B) Weight changes of neonatal rats (C) Naked eye observation of the intestines of neonatal rats (D) HE-stained images of intestinal sections of neonatal rats under light microscope. Magnification × 200 (E) Comparison of intestinal pathology scores of neonatal rats. (F) Intestinal TNF-α concentration in neonatal rats. (G) Intestinal IL-1β concentration in neonatal rats.

### 3. 1- MNA improved NEC intestinal barrier function

Plasma FITC-dextran concentration in the NEC group (Fig 3A) was markedly higher than the control and NEC + 1-MNA groups. Intestinal transmission electron microscopy (Fig 3B) revealed neatly arranged and densely packed microvilli with intact tight junctions (TJs) in the control group. microvilli in the NEC group were sparsely detached, and the structure of TJs was disrupted, whereas 1-MNA intervention restored the structure of microvilli and TJs. Immunohistochemical detection (Fig 3C) of three proteins, Zo-1, Occludin, and Claudin-1, which represent intestinal barrier function, showed significantly decreased expression in the NEC group, which was reversed in the NEC + 1-MNA group. For negative control of immunohistochemistry, refer to S3 File.

### 4. 1-MNA effected on TLR4 and NF-κB pathways

TLR4 and NF-κB pathways play a crucial role in NEC [27]. When TLR4 is activated by pathogenic microorganisms and LPS, it subsequently activates the downstream NF-κB pathway [28]. To explore whether 1-MNA affects the TLR4 and NF-κB pathways, we examined the expression of TLR4, IκBα, and phospho-IκBα by western blot and immunofluorescence. western blot results showed that TLR4 (Figs 4A and 4B) and phospho-IκBα/IκBα (Figs 4A and 4C) were significantly increased in the nec group relative to the normal group and decreased in the nec + 1-MNA group. immunofluorescence results showed that the immunofluorescence signal intensity of TLR4 (Fig 4D) and phospho-IκBα (Fig 4E) was elevated in the nec group. which were alleviated by 1-MNA intervention. Whereas IκBα (Fig 4F) was reduced in the NEC group, 1-MNA administration elevated its expression. This suggests that 1-MNA may play a positive role by inhibiting the TLR4-NF-κB pathway.

### 5. Effect of 1-MNA on the intestinal flora of NEC

The Rarefaction Curve (Fig 5A) flattens out, indicating that the sequencing data volume is sufficient, with additional data expected to yield a limited number of new ASVs. Analysis of the αdiversity index (Fig 5B) showed no significant differences in species diversity among groups (p = 0. 15). The more similar the community compositions of the samples are, the closer they are in the PCA and PCoA plots. PCA (Fig 5C) and PCoA (Fig 5D)plots revealed that the Control and 1-MNA intervention groups had more similar microbial compositions. A graphical representation depicting the relative species abundance at each taxonomic rank (Fig 5E) and a heat map of clustered species abundance at each subclade level (Fig 5F) are shown below. Fusobacteriota and Actinobacteriota were reduced in the NEC group and were partially reverted by the administration of 1-MNA. Campilobacterota was markedly elevated in the NEC group and reverted after administration of 1-MNA intervention. patescibacteria and bacteridota were significantly elevated in the 1-MNA intervention group. Whereas 1-MNA did not seem to have a significant effect on Proteobacteria and Firmicutes in NEC. STAMP differences at the genus level were analyzed (Fig 5G) as follows. Acinetobacter and Escherichia-shigella were elevated in the NEC group, and 1-MNA administration reduced their abundance. However, Lactobacillus was significantly reduced in NEC, and 1-MNA administration did not restore it.

## Disscusion

The metabolite of NNMT, 1-MNA, has been reported to be influential in a range of disease conditions, but its relevance in NEC has not been previously established. Several studies have found that NNMT and its metabolite 1-MNA are elevated in a variety of inflammatory or injurious diseases and exert antifibrotic, anti-inflammatory [14], inhibit NF-κB pathway activity [13] and exert vasoprotective activity [29] Our findings of elevated NNMT and 1-MNA in NEC suggest a potential role for NNMT and 1-MNA in NEC. We further explored the specific mechanisms that 1-MNA may play in NEC using a rat model. Our study demonstrated that 1-MNA intervention resulted in reduced mortality and ameliorated pathological damage in NEC rats. Additionally, 1-MNA decreased the expression of inflammatory factors TNF-α and IL-1β, which is consistent with findings about the attenuation of lipopolysaccharide-induced neuroinflammation by 1-MNA [17].

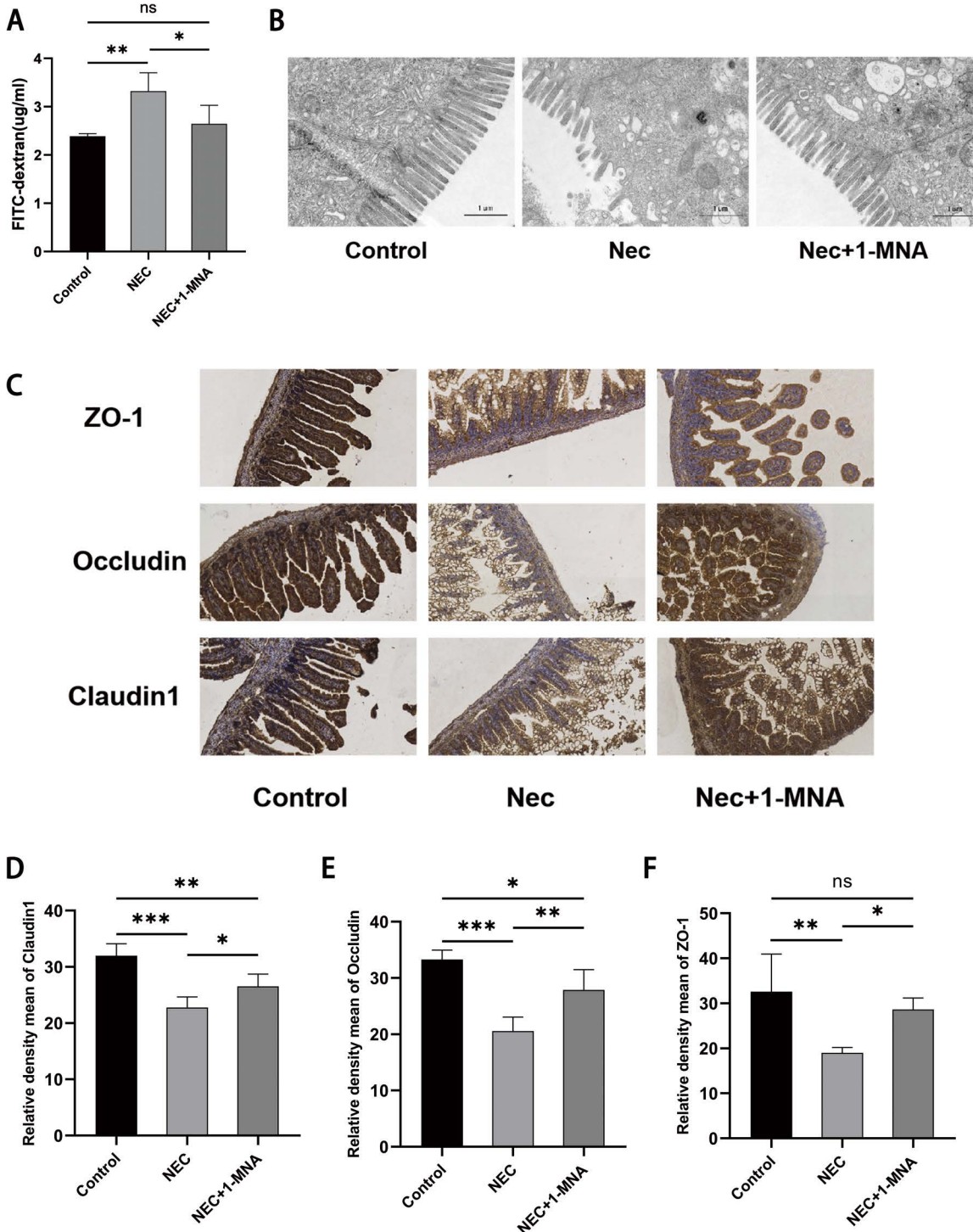

**Fig 3. Effect of 1-MNA on intestinal barrier function in NEC rats** (A) FITC-dextran content in the blood of neonatal rats. (B) Optical electron micros-copy observation of intestinal microvilli and TJ in neonatal rats. magnification × 12, 000 (C) Immunohistochemical detection of the expression intensity of intestinal ZO-1, Occludin and Claudin1 in neonatal rats. (D) Relative expression intensity of Claudin1 in neonatal rat intestine. (E) Relative expression intensity of intestinal Occludin in neonatal rats. (F) Relative expression intensity of intestinal ZO-1 in neonatal rats.

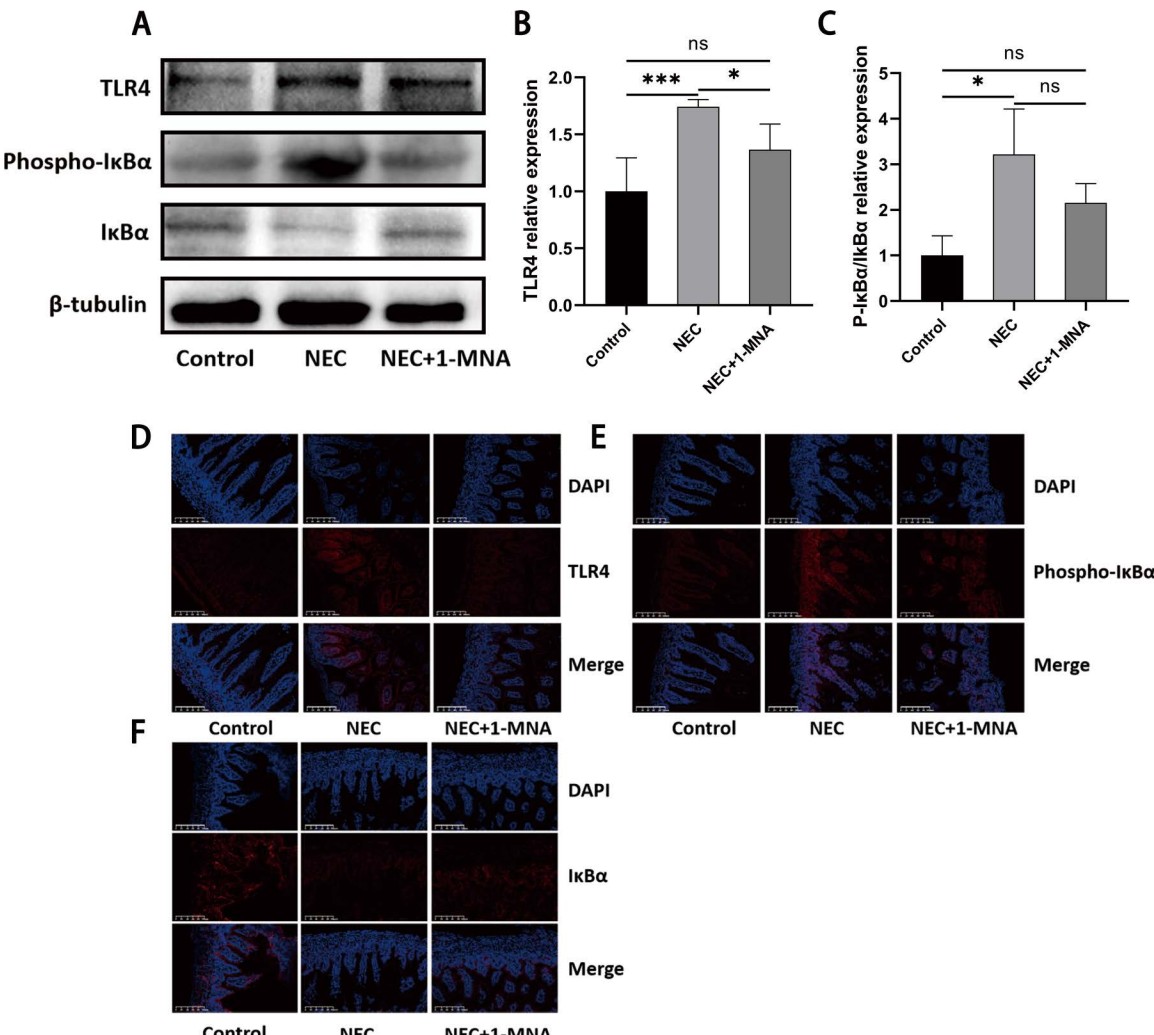

**Fig 4. 1. MNA effected on TLR4 and NF-κB pathways.** (A) western blot detection of protein expression levels of TLR4, phospho-IκBα and IκBα in three groups. (B) Relative expression levels of TLR4 protein in neonatal rats. (C) Relative expression levels of phospho-IκBα/IκBα in neonatal rats. (D) Immunofluorescence detection of TLR4 in neonatal rat intestinal sections. (E) Immunofluorescence detection of phospho-IκBα in neonatal rat intestinal sections. (F) Immunofluorescence detection of IκBα in neonatal rat intestinal sections.

Several studies have reported decreased intestinal TJ protein expression, increased intestinal permeability, and reduced intestinal epithelial barrier function in NEC [30]. The intestinal barrier, situated between the intestinal flora and the host's immune system, serves both immunologic and physical barrier functions [31]. Impaired intestinal barrier function can render the host susceptible to intestinal microbes, potentially leading to systemic inflammatory diseases [32]. This process, known as "bacterial translocation, " may be exacerbated when the mechanisms regulating intestinal barrier repair are disrupted, and thus intestinal barrier function has a significant impact on NEC [33, 34]. Therefore, we further investigated the role of 1-MNA on the NEC intestinal barrier. Electron microscopy [35, 36] and the FITC-dextran permeability assay [37, 38] were utilized to measure intestinal barrier function. TJ proteins such as Occludin, Claudin-1 and ZO-1 are also frequently used as indicators of intestinal barrier function [39, 40]. Our study found sparse shedding of microvilli and disruption of TJ structure in the NEC group under electron microscopy, which was attenuated by the intervention of 1-MNA. The FITC-dextran test also suggested increased intestinal permeability in the NEC group, which was reduced by 1-MNA intervention.

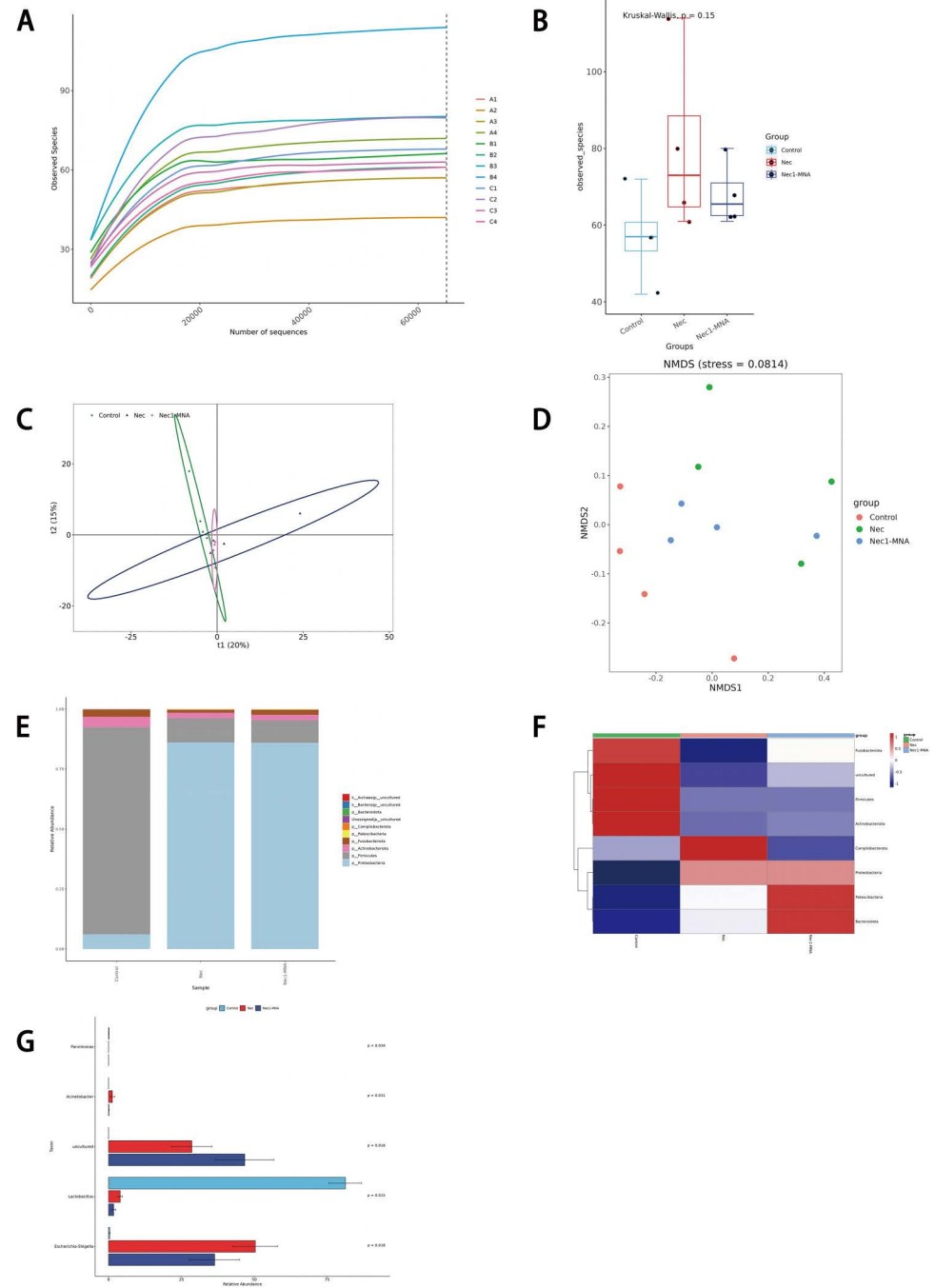

**Fig 5. Effect of 1-MNA on intestinal flora of NEC rats.** (A) Rarefaction Curve of fecal flora of neonatal rats. (B) αdiversity index analysis of fecal flora of neonatal rats. (C) PCA analysis of fecal flora in neonatal rats. (D) PCoA analysis of neonatal rat fecal flora. (E) Histogram of the abundance of neonatal rat fecal flora at the phylum level relative to the species abundance. (F) Heat map of neonatal rat fecal flora phylum level relative to species abundance. (G) STAMP difference analysis plot at the neonatal rat fecal flora genus level.

Immunohistochemical detection of the transmembrane proteins Occludin, Claudin-1, and ZO-1 revealed reduced expression in the NEC group, whereas 1-MNA intervention increased their expression. Collectively, these results suggest that 1-MNA can effectively restore the undermined intestinal barrier function in NEC, thereby slowing disease progression.

   

TLR4 significantly influences the progression of NEC by modulating intestinal epithelial cell damage and repair [41, 42]. Changes in intestinal TLR4 receptors precede histological damage in NEC rats, suggesting that TLR4 may play an initiating role in NEC [43]. The TLR4-NF-κB signaling pathway has been reported to play a crucial role in NEC [27, 44, 45]. Activation of TLR4 induces nuclear translocation of NF-κB and then promotes pro-inflammatory development [46, 47]. Our findings demonstrate that exogenous 1-MNA supplementation downregulates TLR4 expression and inhibits NF-κB activation in NEC rats, as determined by western blot and immunofluorescence assays. This leads to the hypothesis that 1-MNA may relieve intestinal inflammation via the TLR4-NF-κB signaling pathway.

Intestinal dysbiosis is closely associated with NEC, with studies confirming its presence prior to NEC onset [48, 49]. Our study revealed that the 1-MNA intervention group exhibited a bacterial composition more akin to the control group, implying that 1-MNA modulates the intestinal flora in NEC. A systematic review and meta-analysis reported an elevation in Proteobacteria and a decline in Firmicutes and Bacteroidota in NEC babies [50]. Additionally, a reduced abundance of Actinobacteriota in NEC specimens has been noted [51]. In our study, although 1-MNA intervention did not significantly alter Proteobacteria and Firmicutes, it did lead to a significant increase in Bacteroidota and a partial restoration of Actinobacteriota. Furthermore, 1-MNA administration increased the reduced Fusobacteriota abundance in NEC. While studies on Fusobacteriota's role in NEC are limited, our findings suggest a potential link. Pathogenic E. coli and Shigella, common intestinal pathogens, [52, 53] were found in increased abundance in NEC-afflicted children and animal models [54, 55], which is consistent with our study. 1-MNA administration reduced these pathogens' abundance. Acinetobacter, elevated in NEC, was also found to be alleviated by 1-MNA. Although Acinetobacter is a common pathogen, it is not typically a major intestinal pathogen [56]. Thus, 1-MNA appears to play a role in correcting intestinal flora and reducing pathogenic bacteria in NEC.

While our study provides valuable insights, it does not directly elucidate the specific mechanisms of NNMT in NEC. Further research is necessary to explore this relationship. Additionally, as our experiments were conducted on animal models, future cellular studies are required to confirm the safety and delineate the precise mechanisms of 1-MNA action.

## Supporting information

**S1 Table. Antibody dilution concentrations.**
(XLSX)

**S2 File. Representative images for each pathology score.**
(ZIP)

**S3 File. Negative control of immunohistochemistry.**
(ZIP)

**S4 File. The original image with molecular weight labeling for NNMT protein.**
(ZIP)

**S5 File. The original image with molecular weight labeling for TLR4 protein.**
(ZIP)

**S6 File. The original image with molecular weight labeling for Phospho-IκBα protein.**
(ZIP)

**S7 File. The original image with molecular weight labeling for IκBα protein.**
(ZIP)

## Author contributions

**Conceptualization:** Lian Hou, Quan Kang, Junbao Du, Jinxin Li, Qing Luo.

**Formal analysis:** Lian Hou.

**Funding acquisition:** Quan Kang.

**Methodology:** Lian Hou, Quan Kang, Junbao Du, Jinxin Li.

**Project administration:** Quan Kang.

**Resources:** Lian Hou.

**Software:** Lian Hou, Junbao Du.

**Supervision:** Quan Kang, Qing Luo.

**Visualization:** Lian Hou.

**Writing – original draft:** Lian Hou.

**Writing – review & editing:** Lian Hou.

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
