## [Decision Letter · Decision Letter 0]

PONE-D-25-038001-methylnicotinamide attenuated inflammation and regulated flora in Necrotizing enterocolitisPLOS ONE

Dear Dr. KANG,

Thank you for submitting your manuscript to PLOS ONE. After careful consideration, we feel that it has merit but does not fully meet PLOS ONE’s publication criteria as it currently stands. Therefore, we invite you to submit a revised version of the manuscript that addresses the points raised during the review process.

We look forward to receiving your revised manuscript.

Kind regards,

Roberto campagna

Academic Editor

PLOS ONE

Journal Requirements:

3. Thank you for providing details of methods of sacrifice in the online submission form. To comply with PLOS ONE submission requirements, in your Methods section, please provide additional information regarding the experiments involving animals and ensure you have included details on methods of sacrifice.

Additional Editor Comments :

Dear Authors,

the manuscript has been revised and has been found to have some concerns that should be addressed as requested by reviewers in order to consider the manuscript suitable for publication.

Reviewers' comments:

Reviewer's Responses to Questions

**Comments to the Author**

1. Is the manuscript technically sound, and do the data support the conclusions?

Reviewer #1: Yes

Reviewer #2: Yes

2. Has the statistical analysis been performed appropriately and rigorously? 

Reviewer #1: Yes

Reviewer #2: Yes

3. Have the authors made all data underlying the findings in their manuscript fully available?

Reviewer #1: Yes

Reviewer #2: Yes

4. Is the manuscript presented in an intelligible fashion and written in standard English?

Reviewer #1: Yes

Reviewer #2: Yes

5. Review Comments to the Author

Reviewer #1: The manuscript is interesting and generally well written. However, it presents several points that deserve to be improved. In particular:

Introduction:

- It deserves to be pointed out that NEC are an important risk factor for colorectal cancer (see PMID: 28586045)

- "NNMT, a cytoplasmic methyltransferase that catalyzes the N-methylation of NAM to form ........inflammation through its metabolite, 1-MNA13-1" Since this enzyme plays a key role in this manuscript, it deserves to be better introduced. In fact, it plays also a key role in the onset and progression of Gastrointestinal Neoplasms (see PMID: 36139012 )

Materials and methods: authors must provide the product code of all reagents and kits used in order to allow data reproducibility

Histologic assessment of the intestinal tract: a representative image for each score must be provided as supplementary material.

Western blot analysis: i suggest to move the antibodies used in a dedicate table. Moreover, antibodies product code and dilution used must be provided

Western blot images: authors must add the molecular weights in all western blot images

Figure 2D, 4D-F: higher magnifications are needed

Figure 3C: negative control and higher magnifications are necessary

Abbreviations must be written in full length when mentioned for the first time

Reviewer #2: Authors have focused their attention on necrotizing enterocolitis(NEC) a prevalent and destructive illness in neonates. In particular, the paper investigated the role of Nicotinamide N-methyltransferase (NNMT) and its product of enzymatic activity, 1-methylnicotinamide (1-MNA) in the development of this pathology.

The experimental design has been rightly planned and correctly carried out by authors using intestinal tract from both neonatal rats and human newborns. However, the paper requires some revisions before it can be published.

1- The indroduction needs to be better structured. The introduction should be a brief description of the current knowledge about the role of NNMT and its metabolite 1-MNA in the development of patholgies. The introduction should explain the rationale of the study, beginning with the gap in existing literature. In this case, authors mentioned in the introduction the results without explaining why they decided to focus their attention on NEC pathology.

2- The experimental protocol needs improvement. The authors should provide the product codes for the antibodies used but also their dilution in order to make readers able to repeat the experiments

3- The resolution of the figures is very low and needs improvement

4- English needs improvement. There are several grammar mistakes (eg.: 16S rRNA genewas targeted instead of 16S rRNA gene was targeted)

6. PLOS authors have the option to publish the peer review history of their article (what does this mean? ). If published, this will include your full peer review and any attached files.

**Do you want your identity to be public for this peer review?** For information about this choice, including consent withdrawal, please see our Privacy Policy .

Reviewer #1: No

Reviewer #2: No

---

## [Author Response · Author response to Decision Letter 1]

13 Apr 2025

Journal Requirements 1�The manuscript has been modified to conform to the templates as much as possible

Journal Requirements 2�western blot of the original image with molecular weight labeling is supplemented in S4-File.

Journal Requirements 3�Details of animal sacrifice have been added to the methods section.

Journal Requirements 4�A complete ethical statement has been added to the methods section.

Reviewer #1 question 1:Reagent codes have been supplemented in the Materials and methods sections.

Reviewer #1 question 2:Representative images for each score of the intestinal histological assessment have been supplemented in the S2-File.

Reviewer #1 question 3:Product code and dilution of antibody have been added in S1 Table.

Reviewer #1 question 4:Molecular weight in all Western imprinted images has been added in S4-File.

Reviewer #1 question 5:Higher magnification images of Figures 2D and 4D-F have been updated.

Reviewer #1 question 6:The negative control of Figure 3C has been added in the S3-File. The higher magnification figure 3C has been updated.

Reviewer #1 question 7:The full name of the first reference to the acronym has been added to the text.

Reviewer #2 question 1:The introduction explains the validity of the study and explains why we decided to focus on NEC pathology.

Reviewer #2 question 2:The product code of antibody and its dilution have been added in S1 Table.

Reviewer #2 question 3:The image resolution has been improved.

Reviewer #2 question 4:16S rRNA genewas targeted have been changed to16S rRNA gene was targeted.

---

## [Editor Report · Decision Letter 1]

1-methylnicotinamide attenuated inflammation and regulated flora in Necrotizing enterocolitis

PONE-D-25-03800R1

Dear Dr. KANG,

We’re pleased to inform you that your manuscript has been judged scientifically suitable for publication and will be formally accepted for publication once it meets all outstanding technical requirements.

Kind regards,

Roberto campagna

Academic Editor

PLOS ONE
---

## [Editor Report · Acceptance letter]

PONE-D-25-03800R1

PLOS ONE

Dear Dr. Kang,

I'm pleased to inform you that your manuscript has been deemed suitable for publication in PLOS ONE. Congratulations! Your manuscript is now being handed over to our production team.

Kind regards,

on behalf of

Dr. Roberto campagna

Academic Editor

PLOS ONE